# The Complex Interplay between Nevi and Melanoma: Risk Factors and Precursors

**DOI:** 10.3390/ijms24043541

**Published:** 2023-02-10

**Authors:** Rony Shreberk-Hassidim, Stephen M. Ostrowski, David E. Fisher

**Affiliations:** 1Cutaneous Biology Research Center, Department of Dermatology, Massachusetts General Hospital, Harvard Medical School, Charlestown, MA 02129, USA; 2Department of Dermatology, Harvard Medical School, Massachusetts General Hospital, Boston, MA 02114, USA; 3Department of Dermatology, Hadassah Medical Center, The Faculty of Medicine, Hebrew University of Jerusalem, Jerusalem 9112001, Israel

**Keywords:** nevi, melanoma, nevogenesis, melanomagenesis

## Abstract

One effort to combat the rising incidence of malignant melanoma is focused on early detection by the clinical and dermoscopic screening of melanocytic nevi. However, the interaction between nevi, which are congenital or acquired benign melanocytic proliferations, and melanoma is still enigmatic. On the one hand, the majority of melanomas are thought to form de novo, as only a third of primary melanomas are associated with a histologically identifiable nevus precursor. On the other hand, an increased number of melanocytic nevi is a strong risk factor for developing melanoma, including melanomas that do not derive from nevi. The formation of nevi is modulated by diverse factors, including pigmentation, genetic risk factors, and environmental sun exposure. While the molecular alterations that occur during the progression of a nevus to melanoma have been well characterized, many unanswered questions remain surrounding the process of nevus to melanoma evolution. In this review, we discuss clinical, histological, molecular, and genetic factors that influence nevus formation and progression to melanoma.

## 1. Introduction

Malignant melanoma is an aggressive type of skin cancer that originates from melanocytes, the cells responsible for skin pigmentation. In its advanced stages, melanoma has a poor prognosis; therefore, over the years, risk factors and early identification of suspected precursor lesions for melanoma formation have been explored to improve its diagnosis at early stages prior to metastatic spread [1]. A major focus has been given to the melanocytic nevus, which is defined as a benign skin lesion composed of clustered melanocytes [2]. Clinically, increased numbers of melanocytic nevi and the presence of atypical nevi have been recognized as risk factors for the development of melanoma [3]. In a meta-analysis on nevi as risk factors for melanoma, the highest risk (about 7-fold) for melanoma was observed in individuals with more than 100 nevi [3]. The existence of several atypical nevi was correlated with a six-fold higher risk for melanoma formation in comparison with the absence of atypical nevi [3]. This dose-dependent correlation between nevus counts and melanoma was thought to imply that nevi may represent precursor lesions in the evolution of melanoma. However, the risk of a single nevus progressing to melanoma is very low (less than 0.0005% annual risk), so most nevi will remain stable and not transform into melanoma [4].

When examining nevus remnants in histology samples of melanoma, only a third of melanomas are nevus-associated (nevus-associated melanoma, NAM), while most melanomas are thought to form de novo [5]. Even though the reported rate of NAM varies, the most commonly reported rate is 20–30% [5,6]. A possible explanation for the underestimation of NAM’s low rate is that in thicker melanoma, the nevus component cannot be detected since it may be obscured or destroyed by the tumor component [7]. Small, subclinical nevus nests have been reported to occur commonly in human skin specimens, and it is possible that these “micronevi” may serve as melanoma precursors and be so small as to evade detection during the analysis of melanomas [8,9]. On the other hand, some authors have suggested that NAM is overdiagnosed and that many nevus remnants represent areas of melanoma with nevoid differentiation [10].

Current evidence suggests a clinical and biological link between nevus formation and the development of melanoma. However, the differences between NAM and de novo melanoma formation remain poorly understood. Here we review the current literature on characteristics of NAM versus melanoma de novo and factors that may impact their differences.

## 2. Current Knowledge of the Classification of Melanoma Formation Pathways

The World Health Organization (WHO) classified, in 2018, nine different evolutionary pathways for melanoma formation, based on genetic alterations, potential precursor lesions, and degree of chronic sun damage (CSD). These pathways include: Low-CSD melanoma, such as superficial spreading melanoma (SSM) that may arise from a nevus or dysplastic nevus; High-CSD melanoma, such as lentigo maligna melanoma that may originate from melanoma in situ rather than benign precursors; desmoplastic melanoma, which does not have a known benign precursor; Spitz melanoma that may have a Spitz nevus as a precursor; acral melanoma, with a minority arising from nevi; mucosal melanoma that does not have a known benign precursor; melanoma arising from congenital nevus; melanoma arising in a blue nevus; uveal melanoma that may have uveal nevus as precursor; Nodular melanoma can be classified as part of either low-CSD or high-CSD in the WHO classification, depending on the level of CSD [11,12].

Congenital nevi are a classic example of NAM. They are typically present at birth or in the first years of life. This type of nevus is usually caused by an activating mutation in the MAPK pathway, mainly the *NRAS* gene [13]. Large-size and giant congenital nevi (>20 cm) exhibit a significantly increased risk for melanoma formation, with an estimated lifetime risk of 10–15%, but likely requiring further mutational events (other than *NRAS*) and/or genomic alterations in order to progress [14,15]. A definitive example of melanoma arising de novo without a benign precursor lesion is lentigo maligna melanoma (LMM). It originates from melanoma in situ lesions (lentigo maligna) which are slowly expanding and only rarely progress into invasive melanoma [16]. It is usually found in older patients on sun-damaged skin, such as the head and neck areas [17].

WHO classification emphasizes the distinction between melanoma developing de novo versus melanoma arising from a benign precursor lesion, namely NAM. Importantly, while NAM and melanoma de novo broadly represent separate clinical, histological, and molecular entities, there is an overlap. For example, while LMM and nodular melanoma most commonly occur de novo, a small percentage can occur in association with a nevus precursor [18,19]. Conversely, while SSM is enriched in NAMs, about one-third of SSM cases occur de novo [18]. Survival (when normalized to Breslow depth, a key parameter for invasion and risk of relapse) is similar in NAM and de novo melanoma [20]; to date, no reports have examined differences between immunotherapy response in metastatic melanoma patients whose primary melanoma was de novo versus nevus-associated [20]. Elucidation of mechanisms underpinning nevus-associated versus de novo melanomagenesis may be helpful for better understanding melanoma risk and developing improved prevention strategies.

## 3. Insights on the Clinicopathological Characteristics of Nevus-Associated Melanoma versus Melanoma De Novo

The evolution of melanoma from precursor lesions, including benign melanocytic nevi, has been carefully and convincingly shown to correlate with the stepwise progression of genetic modifications [21]. The most common first step in the formation of a precursor lesion is the activation of the MAPK (mitogen-activated protein kinase) pathway by activating driver mutation in *BRAF* or *NRAS* genes, inducing the formation of a benign nevus. It is unknown whether BRAF^V600E^ mutation is fully sufficient for nevus formation in vivo [22]. In vitro, BRAF^V600E^ activation leads to rapid senescence of primary human melanocytes without a period of growth advantage [23]. It is possible that, in vivo, there is prolonged latency between the proliferation and senescence induced by BRAF^V600E^. This is supported by the findings that BRAF^V600E^ activation in zebrafish and mice leads to the production of nevus-like lesions [24,25]. An alternative hypothesis suggests that more rapid growth arrest, analogous to that observed in vitro, also occurs in vivo but results in arrested melanocytes or melanocyte proliferations that are too small to detect clinically. In support of this hypothesis, a common clinical finding is a broad variability in nevus counts, even amongst patients with similar skin types and ultraviolet (UV) exposure [26]. This suggests that the limiting factor is not BRAF^V600E^ mutation itself, but instead, other genetic factors that may constrain or promote the expansion of BRAF^V600E^ mutant melanocytes to clinically appreciable nevi. Similarly, “eruptive” BRAF^V600E^ positive nevi have been described in patients on immunosuppressive medications, suggesting that BRAF^V600E^ mutant melanocytes were present in the skin but restrained by immune-dependent mechanisms [27].

The growth of a *BRAF* or *NRAS* mutant nevus is limited by cell-cycle arrest and cellular senescence, which may be interrupted by genetic and genomic disruption of tumor suppressor genes to allow the progression of the lesion to melanoma. These accumulating mutations include TERT (telomerase reverse transcriptase) promoter mutations, components of the SWI/SNF (switch/sucrose non-fermentable or BAF) chromatin remodeling complex and loss of *CDKN2A* which are early events noted along the trajectory of nevus to melanoma transition [21]. Disruption of the p53 pathway and/or activation of the PI3K pathway are later steps in the progression to invasive melanoma [21]. These findings support linear evolution from nevus to melanoma. Linear evolution is also suggested by the existence of nevus remnants in a fraction of histological samples of melanoma, such as in NAM.

Key clinicopathological differences have been identified between NAM and melanoma de novo (Table 1), even when considering only thin melanomas (to avoid bias of possible destruction of nevus precursor in thicker melanomas) [28]. In many studies, including a recent meta-analysis by Pampena et al. [5], the clinical features associated with NAMs included younger age at diagnosis, increased total nevus count and anatomic sites with intermittent sun exposure, such as the trunk [28,29,30]. In most reports, NAMs are associated with acquired nevi, specifically intradermal type [5]. Melanoma de novo was correlated with older age at diagnosis and locations with cumulative sun damage, such as the extremities and head/neck [31,32].

Regarding dissimilarities in histopathology, NAM was more commonly associated with the SSM histological subtype, thinner Breslow, regression, absence of solar elastosis and absence of ulceration [6,33]. In contrast, melanoma de novo more typically includes nodular melanoma subtype, thicker Breslow, presence of ulceration, solar elastosis, and absence of regression [28,32]. In most studies, no difference in survival was found between NAM and melanoma de novo [32,34,35]. However, this question is especially challenging to explore since melanoma de novo has been linked to thicker Breslow and ulceration, both poor prognosis factors of melanoma.

Since the presence vs. absence of associated nevi in early melanoma represents distinct clinicopathological characteristics, it is plausible that they truly represent distinct pathways of melanomagenesis. Further studies will be reviewed to better understand the mechanisms involved in the processes generating these alternative routes to tumor formation.

## 4. Factors Modulating the Nevus to Melanoma Evolution

### 4.1. Genetic Factors

Currently, several genetic variants have been identified to play key roles in the formation of both nevus and melanoma, suggesting that there are shared genetic pathways that drive the growth of both lesions. These ‘nevi and melanoma’ genes include *BRAF*, *CDKN2A*, *MITF*, some telomere length maintenance genes, and *IRF4* [36,37,38]. We will elaborate more on each one of these genes to describe their impact on the interaction between nevogenesis and melanomagenesis. Figure 1 illustrates a proposed model of genetic alterations that cause the nevus-to-melanoma transformation.

Importantly, when this has been examined, genetic factors that increase melanoma and nevus risk increase the risk of both de novo and NAM. For example, *CDKN2A* germline mutation dramatically increases both melanoma risk and risk of multiple dysplastic nevi, but the percentage of NAM in *CDKN2A* germline patients is similar to the general population [39,40]. Similarly, patients with high mole counts have increased melanoma risk, but this higher risk includes both nevus-associated and de novo melanomas [41]. This suggests that the biological processes that regulate nevogenesis and melanomagenesis are linked.

However, there are melanoma-specific and nevus-specific polymorphisms. KIT Ligand (KITLG, also known as stem cell factor/SCF) is an important melanocyte signaling molecule and is encoded by the *KITLG* gene, which is the ligand for the melanocyte signaling receptor Kit. *KITLG* regulates skin pigmentation and has been implicated as a nevus risk gene but is not known to influence melanoma risk [42]. Conversely, many skin pigmentation loci (*TYR, OCA2, ASIP*) and most telomerase components influence melanoma risk but do not apparently impact nevus count [42]. Some telomere length maintenance genes (*TERC, OBFC1*) regulate both nevus count and melanoma risk, while the majority regulate melanoma risk only (such as *TERT, DKK2, NAF1*). Polymorphisms in *IRF* (discussed in detail below) offer a unique exception as *IRF4* polymorphisms lead to divergence of nevus associated and de novo melanoma risk.

#### 4.1.1. BRAF, NRAS and TERT

Activating mutations in the *BRAF* proto-oncogene are of special interest since they are found in more than 80% of benign nevi [43]. These mutations typically involve the substitution of glutamic acid for valine at codon 600 (*V600E*), which leads to the constitutive activation of BRAF and downstream signaling pathways (MAPK pathway), resulting in increased cell proliferation [44]. *BRAF^V600E^* mutation is also found in more than half of all cutaneous melanomas [45]. Since benign nevi are small and stable over time, it is believed that their growth is limited by an oncogene-induced senescent state, suggesting that the *BRAF^V600E^* mutation alone, as aforementioned, is not sufficient to transform normal melanocytes into melanoma cells [46]. The *NRAS* gene, which is an additional stimulator of the MAPK pathway, is mutated in 15–20% of melanomas [47]. It is infrequently mutated in acquired melanocytic nevi (<5%) but is the predominant mutated driver oncogene in congenital nevi; 95% of large congenital nevi are NRAS mutant, and these lesions have a higher propensity for melanoma transformation [13,14,48,49]. Activation of the MAPK signaling pathway was also demonstrated as a result of loss-of-function mutations in the Neurofibromin (NF1) gene. It is a tumor suppressor gene that facilitates the inactivation of RAS, and its loss was found in 10–15% of melanomas [50,51].

The *TERT* gene encodes a component of telomerase; thus, when this gene is upregulated, it enables the melanocytes to evade senescence and escape telomere-dependent growth arrest [52]. It may explain why *TERT* mutations are crucial for the malignant transformation of melanocytes and the initial steps of melanoma development. An example of the essential impact of *TERT* mutation on the early stages of melanomagenesis was demonstrated by Bastian and his group, who suggested a stepwise accumulation of mutations during the process of nevus to melanoma transformation. They found that almost 80% of intermediate lesions and melanoma in situ carry *TERT* promoter mutation [21].

It has been proposed that the chronological order of the mutations in *BRAF^V600E^* and *TERT* has an influence on the model of melanoma evolution. If the initial genetic step is a mutation in the *BRAF^V600E^*, which is relatively common, the consequence is a benign acquired nevus. Subsequent modifications in the *TERT* promoter and other genes lead to the progression of nevus into melanoma. Nevertheless, if the mutation in the *TERT* promotor (or in other genes regulating telomerase) occurs first, it may result in neoplastic modifications of epidermal melanocytes. At first, the lesion may be latent, but later alterations in proliferation genes, such as *BRAF*, lead to its progression to melanoma in situ [10]. This model implies that NAM is associated with an initial mutation in *BRAF^V600E^*, while in melanoma de novo, a primary step is an alteration in the *TERT* promoter. The *BRAF^V600E^* mutation is more prevalent in NAM and, as expected, is also associated with clinicopathological characteristics of NAM, including truncal location and younger age [53].

#### 4.1.2. CDKN2A

The somatic deletion and mutation of the *CDKN2A* locus are among the most common genetic alterations in melanoma, responsible for the progression from precursor lesions to invasive melanoma [38]. This locus encodes two tumor suppressor gene products, including the cell-cycle inhibitor p16 (INK4A) and p14 (ARF), which prevents the degradation of p53 [54,55]. The biallelic inactivation of this locus is associated with the invasive behavior of melanoma, which was attributed to a disruption in its cell cycle suppression effects. Recently, a mechanism by which deletion of *CDKN2A* results in melanoma progression was described. It was shown that loss of p16 leads to activation of the transcription factor BRN2, mediated by retinoblastoma protein (RB1) and the E2F transcriptional pathways. Increased levels of BRN2 were related to enhanced motility and invasion abilities of melanocytes. It correlated with the loss of p16 (INK4A) together with increased expression of BRN2 in invasive melanoma versus melanoma in situ samples [56].

The most common alteration in familial melanoma involves the *CDKN2A* locus, which is altered in about 20–40% of familial melanoma cases [27,28]. In family members carrying *CDKN2A* mutation, a phenotype of multiple clinically atypical melanocytic nevi was recognized, often entitled “dysplastic nevus syndrome.” These nevi are often larger in size, with variable colors and irregular borders, and may demonstrate atypical features in histology [57]. It is not clear if nevi with clinical and histological atypical characteristics have an increased propensity to progress to melanoma. Also, in a case-control study comparing the histopathology of *CDKN2A*-mutated with sporadic melanomas, no difference was found in the presence of an adjacent nevus [39].

Therefore, it is still challenging to correlate *CDKN2A* mutations and a specific pattern of melanoma evolution. It may be further complicated by the relatively high prevalence of loss of this locus in sporadic and hereditary melanomas. However, it is likely that loss of this locus is correlated with acquiring invasive ability of the tumor, a feature which is suppressed in melanoma in situ and other precursor lesions.

#### 4.1.3. IRF4

*IRF4* encodes the interferon regulatory factor 4, which plays a role in the differentiation of immune cells. This transcription factor has also been used as a marker in histologic samples of certain cancers, including hematologic malignancies and melanoma [58]. This gene is related to skin pigmentation by modulation of tyrosinase expression, a key enzyme in the pigmentation pathway [59].

Polymorphism in a specific variant of the *IRF4* gene, the rs12203592 variant, was associated with different features of melanoma histologic samples. The variant rs12203592*T was associated with melanoma and solar elastosis, a histologic marker for sun exposure. While this variant was inversely correlated with the existence of nevus precursor, the rs12203592*C variant was positively associated with the presence of nevus in the melanoma samples [60]. The rs12203592*T variant was shown to be related to additional features, including higher Breslow thickness and melanoma-specific survival and a lower likelihood of harboring the *BRAF^V600E^* mutation [61,62]. In relation to nevus counts, the rs12203592*T variant was more prevalent in adults with low nevus counts and high freckling scores, while, in adolescents, the opposite association was found [63]. Therefore, these data may suggest that the IRF4 rs12203592*T variant is more likely to be correlated with melanoma de novo; however, more study is warranted. This offers a provocative example in which different polymorphisms in a single gene might offer differential risks of nevi associated and de novo (high-CSD) melanoma pathways. Mechanistic understanding of these differential effects could offer important insights into the pathogenesis of NAM and de novo melanoma.

#### 4.1.4. Other Genetic Alterations

The transcription factor microphthalmia (MITF) is a melanocyte lineage-specific factor that is required for proper melanocyte development. It can be amplified or dysregulated and thereby function as a melanoma oncogene. A rare germline *E318K* polymorphism has been shown to alter MITF activity by disrupting sumoylation and consequently increasing the risk for melanoma [64,65]. Melanoma patients that were *E318K* carriers had a statistically significantly higher number of nevi in comparison with MITF wild-type patients; interestingly, these nevi had similar clinical and dermoscopic features as MITF wild-type patients [66].

NAM and melanoma de novo exhibit differences in relation to chronic sun damage (CSD). As mentioned above, NAM is more frequently associated with intermittent sun exposure, while melanoma de novo is associated with chronic sun exposure. The pattern of sun exposure has an influence on diverse genetic alterations leading to melanomagenesis [43]. UV-induced DNA damage is thought to more frequently involve the *TP53* tumor suppressor gene, which is more commonly mutated in CSD-associated-melanoma [38,45]. Interestingly, in immunohistochemistry analysis of melanomas with adjacent nevi, it was demonstrated that the tumor component expressed p53 at high levels, while in the nevus component, the levels were low [67]. These data suggest that *TP53* mutations occur late in melanoma evolution, as was previously described by Shain et al. [21].

*PTEN*, a tumor suppressor, and *BRAF* have been commonly described in cutaneous melanomas not related to CSD. *PTEN* loss is particularly associated with *BRAF^V600E^*-mutated melanomas, which may imply a synergistic effect of both mutations in inducing melanoma formation [68]. Since after *PTEN* loss, melanoma rapidly forms in melanocyte-targeted *BRAF*-mutated mouse models, it was suggested that *PTEN* is a pivotal factor in maintaining the growth arrest of nevi. The inactivation of *PTEN* in *BRAF*-mutated nevus may represent a leading step in the complex process of nevus to melanoma transformation, thus NAM formation [69,70].

To summarize, key tumor suppressor genes, such as *TP53* and *PTEN,* were previously described in relation to melanoma development. *TP53* loss is more common in CSD-melanomas, usually not related to a nevus precursor. However, *PTEN* inactivation may result in the transformation to melanoma in *BRAF*-mutated nevi, suggesting it may be important in the pathogenesis of NAM.

### 4.2. Pigment Background

Melanocytes serve a key role in the protection of skin cells from UV-induced DNA damage to prevent carcinogenesis. Melanin exists in two chemical forms, black/brown eumelanin and red/orange pheomelanin. After being synthesized within melanocytes, melanin-containing melanosomes are subsequently transferred to keratinocytes [1]. Eumelanin/pheomelanin balance is regulated by the melanocortin-1-receptor (MC1R)/ MITF pathway. *MC1R* loss-of-function polymorphisms result in decreased flux through the MITF pathway with a preference for pheomelanin synthesis. These *MC1R* variants are common in fair-skinned European populations and are strongly associated with red hair, fair skin, and increased melanoma risk [71,72]. Thus, increased melanoma risk in this population may, in part, be caused by the inherent decreased UV shielding effects of pheomelanin. Prior data have also demonstrated the existence of a UV-independent mechanism of increased melanoma risk induced by the pheomelanin synthetic pathway [73,74,75]. In transgenic mouse studies, it was shown that increased melanoma risk caused by *MC1R* inactivation was dependent upon pheomelanin production even in the absence of UV, and this risk was reversed by ablation of all pigmentation (introduction of an albinism mutation). It was suggested that the mechanism by which pheomelanin production promoted melanoma development was increased reactive oxygen damage [75]

As mentioned above, increased nevi count represents an independent risk factor associated with melanoma formation. It is known that individuals with fair skin and light color of the eyes have higher nevi counts and susceptibility for melanoma development, attributed in part to their propensity for sunburns. Surprisingly, red-haired individuals (harboring *MC1R* polymorphism), which represent the skin phototype with the highest risk for melanoma development, have low nevus counts [76]. Interestingly, the rate of NAM in red-haired individuals is thought to be very similar to the rate of NAM in non-red-haired individuals [30]. There are several possible explanations for the gap between increased melanoma risk and low nevi counts in red-haired individuals. One hypothesis is that these individuals do not have numerous nevi, but each nevus has a higher propensity for melanoma transformation. Pheomelanin-dependent oxidative stress may promote nevus-to-melanoma formation, and detailed analysis in transgenic mouse models may elucidate the role of *MC1R*/pheomelanin in nevus formation and nevus-to-melanoma transition.

## 5. Conclusions and Future Implications

Abundant literature suggests that there are several patterns for nevogenesis and melanoma evolution, depending on phenotypic features and genetic alterations. In recent years, studies have begun to focus on the characterization of two distinct patterns of melanoma formation, NAM and melanoma de novo. However, these data have not yet been translated into either deep mechanistic understanding or potential clinical applications. We propose a model for differentiating between these classes of melanomas and also suggest some screening and therapeutic considerations.

NAM is associated with the “nevus-prone” phenotype, which is characterized by younger age at diagnosis, multiple nevi, relationship to low CSD and *BRAF^V600E^* mutation. This population may require dermatological screening starting at a younger age for early detection of nevi transforming into melanoma. As these melanomas more frequently harbor *BRAF^V600E^* mutation, treatments with BRAF-targeted small molecules may also offer advantages (potentially in the adjuvant or advanced stages).

Melanoma de novo is associated with the “non-nevus” phenotype, which is characterized by older age at diagnosis in relation to high CSD and *TP53* mutations. While all populations should receive meticulous education regarding sun protection (which may impact melanoma risk many years later), the melanoma de novo population may be more likely to develop new skin lesions rather than a transformation from pre-existing nevi (although not with perfect predictability). Immunotherapy may be successful in treating such melanomas if they develop within high-CSD skin and may therefore accumulate a high incidence of UV-induced DNA damage, which generates neo-antigens amenable to immunotherapy targeting.

It remains to be determined whether these patterns of NAM vs. de novo melanoma indeed reflect varied therapeutic responsiveness, as suggested by these considerations. Nonetheless, it is hoped that greater knowledge of the mechanistic underpinnings of these pathways for melanoma formation may eventually offer improved prevention, early detection, and therapeutic interventions aimed at decreasing mortality from this life-threatening disease.

## Figures and Tables

**Figure 1 ijms-24-03541-f001:**
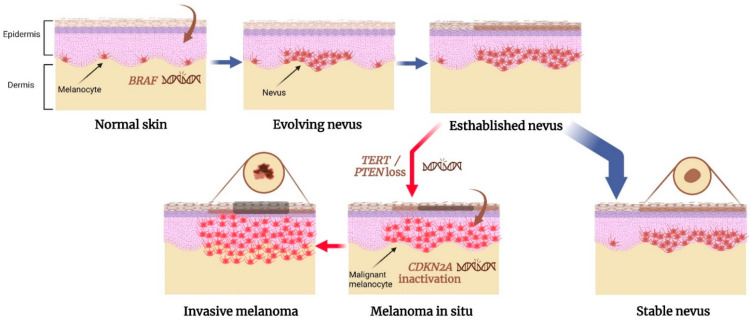
A proposed model for genetic alterations involved in nevus to melanoma transformation. *BRAF^V600E^* mutation is a common initial step in the formation of acquired nevus, resulting in the benign proliferation of melanocytes in the skin. The nevus reaches a senescent state and remains stable. However, if a mutation in a telomerase-regulating gene or loss of a tumor suppressor gene occurs, the melanocytes evade senescence. It may lead to the transformation of the nevus into an intermediate lesion or even melanoma in situ. Inactivation of both *CDKN2A* genes may be involved in the progression to invasive melanoma. Created with BioRender.com.

**Table 1 ijms-24-03541-t001:** Comparison between associated characteristics of nevus-associated melanoma and melanoma de novo.

	Nevus-Associated Melanoma	Melanoma De Novo
Clinical		
Age at diagnosis	Young	Old
Total body nevi count	High	Low
Body site	Trunk	Head/neck
Sun exposure	Intermittent sun exposure	Chronic sun exposure
Histological		
	Superficial spreading subtype	Nodular melanoma subtype
	Thin Breslow	Thick Breslow
	Regression	Absence of regression
	Absence of ulceration	Ulceration
Initial genetic alteration	*BRAF V600E*	*TERT* and other telomerase-related genes
Other genetic alterations	*PTEN* inactivation	*TP53* inactivation

## Data Availability

Data sharing not applicable.

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
