# Peer review of "The Complex Interplay between Nevi and Melanoma: Risk Factors and Precursors"

_ijms, 2023, doi:10.3390/ijms24043541_

Round 1
Reviewer 1 Report
This is a timely and well-written review on a topic that received much less attention. Overall, the authors presented an easily readable review of the state of the field. There are few minor issues that need attention:
1. The authors make no mention of NF1 mutations. A sentence or two would be appropriate to include.
2. On Page 2, lines 95-99, while discussing the evolution of melanoma from adjacent precursor, the authors state that the the first step is activation of the MAPK pathway, by activating driver mutation in BRAF or NRAS genes. To readers that are unfamiliar with the field, this could cause confusion in that BRAF/NRAS mutations occur in lesions that are already considered adjacent precursors.
3. On Page 3, line 124, there is an extra floating letter- s
4. Page 5, line 167- there is an incomplete sentence/floating words "TERT, telomerase reverse transcriptase."
Author Response
Reviewer 1:
This is a timely and well-written review on a topic that received much less attention. Overall, the authors presented an easily readable review of the state of the field. There are few minor issues that need attention:
1. The authors make no mention of NF1 mutations. A sentence or two would be appropriate to include.
Thank you for this valuable comment. Information regarding NF1 mutation was added to the manuscript.
2. On Page 2, lines 95-99, while discussing the evolution of melanoma from adjacent precursor, the authors state that the first step is activation of the MAPK pathway, by activating driver mutation in BRAF or NRAS genes. To readers that are unfamiliar with the field, this could cause confusion in that BRAF/NRAS mutations occur in lesions that are already considered adjacent precursors.
Thank you for pointing out this important comment. We aimed to describe the genetic stepwise evolution of melanoma from a nevus precursor. The first step in this process is the BRAF/NRAS mutations, which result in nevus formation. We have edited the text to remove the unintended ambiguity. The changes are near the bottom of page 2 (line 95) and on the top of page 3 (lines 97 and 98). We believe this should appropriately clarify the meaning.
3. On Page 3, line 124, there is an extra floating letter- s
Thank you. It was deleted.
4. Page 5, line 167- there is an incomplete sentence/floating words "TERT, telomerase reverse transcriptase."
We appreciate this correction. The incomplete sentence is part of the figure legend, to explain what TERT in the figure stands for. To avoid confusion, it was deleted.
Reviewer 2 Report
I really appreciate the opportunity to review this manuscript. Although the manuscript is very well-written and the concept and the way of integration were reasonable, but, I have some concerns regarding this manuscript.
1. This manuscript is well written but not novel. Very similar review articles have been published, and I don’t see what this manuscript adds to existing literature. If systematic reviews or meta-analyses are added on this topic, it will be a more meaningful manuscript that organizes existing review papers more systemically and provides a new perspective.
2. The discussion to publish or not depends on the policy of the Editor.
Author Response
Reviewer 2:
I really appreciate the opportunity to review this manuscript. Although the manuscript is very well-written and the concept and the way of integration were reasonable, but I have some concerns regarding this manuscript.
1. This manuscript is well written but not novel. Very similar review articles have been published, and I don’t see what this manuscript adds to existing literature. If systematic reviews or meta-analyses are added on this topic, it will be a more meaningful manuscript that organizes existing review papers more systemically and provides a new perspective.
We appreciate your feedback. We respectfully disagree and believe that our manuscript does indeed add valuably to the existing literature by providing a comprehensive review of the current literature on both melanoma de novo and nevus-associated melanoma. With the aim of novelty, our manuscript included discussion of several topics that have not been highlighted in recent reviews: 1) The sufficiency of BRAFV600E in inducing nevogenesis; 2) The role of the MC1R pathway in nevogenesis; 3) Role of MITF germline mutations in promoting nevogenesis; and 4) The existence of “micronevi” and whether they may serve as a precursor for melanoma. We understand that further systematic reviews or meta-analyses may add further value to the manuscript, and we will consider incorporating them in future publications.
Round 2
Reviewer 2 Report
There is no objection to publishing because the corrections I suggested were well reflected.
However, as stated in the previous review comment, the manuscript is well written but not novel. The decision to publish or not depends on the policy of the Editor.